# Banking 4.0: Artificial Intelligence (AI) in Banking Industry & Consumer's Perspective

**Umara Noreen** [1] **, Attayah Shafique** [2,*] **, Zaheer Ahmed** [3] **and Muhammad Ashfaq** [4]

[1] College of Business Administration, Prince Sultan University, Riyadh 11586, Saudi Arabia
[2] Department of Communication and Management Sciences, Pakistan Institute of Engineering and Applied Sciences, Islamabad 45650, Pakistan
[3] Faculty of Business Administration, Iqra University, Karachi 75500, Pakistan
[4] Department of Business & Management, IU International University of Applied Sciences, 53604 Bad Honnef, Germany
[*] Correspondence: attayah.shafique@pieas.edu.pk or attayahshafique@gmail.com

**Abstract:** The simulation of human intelligence in machines, called Artificial intelligence, has risen, and plays an important role in the new banking era. The present study aims to discuss the consumer's perspective on artificial intelligence's adoption in Asian countries. The questionnaire was developed and distributed to collect data from five Asian countries (Pakistan, China, Iran, Saudi Arabia, and Thailand). The total useable responses were 799. The results showed that the factors (awareness, attitude, subjective norms, perceived usefulness, and knowledge of artificial intelligence technology) had a significant and positive relationship with the intention to adopt AI in the banking sector. However, perceived risk shows a negative but significant relationship with the intentions to adopt AI. Overall, the findings of this study will be a worthy insight for making strategic decision-making in the banking industry. This will enable the banking management to build a strategy to increase the trust of consumers, which will help them to overcome risks and give them confidence in using digital technology while making transactions. The banking sector also focuses on innovative AI technologies to improve customer services as well as overall growth by generating more revenue.

**Keywords:** banking sector; awareness; knowledge of AI technology; consumers

## 1. Introduction

The concept of Artificial intelligence refers to a broad field of science encompassing not only computer science but psychology, philosophy, linguistics, and other areas [1]. AI offers huge benefits to the global economy and financial services industry. According to a recent study [2], AI can create additional value of up to USD 1 trillion each year for the global banking industry. According to [3], the global financial services industry is expected to reach USD 28.529 trillion by 2025–2030 at a compound annual growth rate (CAGR) of 6%. This is mainly because of the substantial use of AI in the rearrangements of banking operations, particularly after recovering from COVID-19.

The evolution of the banking industry started from Banking 1.0, which is based on traditional and historical banking to Banking 4.0, which comprises advanced technology used in different areas in banks, including the use of AI technologies. Banks have been using new and cutting-edge technologies to remain relevant and stay ahead of the competition. For instance, by using automated teller machines (ATMs) during the 1960s [4], which were introduced under Banking 2.0 and initiated by Barclays Bank. The rapid advances in AI technologies facilitated the reduction in the cost of data processing, storage, and faster connectivity in 2017 onwards and are known as Banking 4.0.

The attractiveness of AI-based financial inclusions at lower operating costs witnessed an accelerated shift in banking services. The commercial licensing initiatives of electronic money institutes (EMI) increased the EMIs three times to 40.8 million [5]. Banks have made

a huge quantum of investments in developing AI infrastructures and innovative financial inclusions to cater to the growing demand for financial products in the economy. This uptake of AI recorded a 7.6% growth rate on quarterly bases spurred by 7.6 million banking users [6]. It is also observed that the banking sector that uses artificial intelligence generates higher revenues such as JPMorgan Chase, CitiBank, Wells Fargo, Barclays Bank Plc, Capital One, etc.

Despite the attractiveness and implications of AI technologies in the banking sector, there are several hindrances in terms of the adoption of AI technology in the banking sector. This can be attributed mainly due to a lack of knowledge about AI technology among consumers. Previous research has explored this aspect but in generalized terms. However, more studies are needed in terms of AI technological competence. Therefore, this research focuses on AI knowledge specifically and not on basic technology use, such as internet and computer proficiency as suggested in previous studies [7]. Additionally, this research has gathered data from five Asian countries with a larger sample size than suggested in previous research [7,8].

Today's banking sector needs to be updated in terms of the consumers' perception of financial technologies, especially on the role of artificial intelligence. Therefore, to be at par with the banking industry, it is important to understand the consumer's knowledge about AI technology specifically.

This exploratory research from the consumer's perspective will help the banking industry to understand the user's perspective in terms of their awareness, perception, trust, and norms towards this transformation. This study will enlighten the banking industry's policymakers to devise a strategic plan for AI adoption strategies.

The rest of the paper is as follows: the next section is a literature review followed by the methodology. Section four is about the results and discussion, and the final section is about the conclusion, limitations, and future research.

## 2. Literature Review

### 2.1. AI Application Areas in the Banking Sector

The application of AI has been found in multiple fields, including government payments, healthcare, online trading businesses, logistics, the financial industry, etc. [9]. AI can help banks both manage their financial services and engage customers while offering personalized products [10]. It has the flexibility to fulfill the needs of organizations of all sizes, from small to large. AI is based on the computation of mathematical complex algorithms that communicate at a high-speed base on given logical conditions through computer systems. According to [11], AI-based digital financial services are more efficient and faster than the traditional methods of performing various financial computational tasks in banking operations. Moreover, with the inherent nature of AI and the competitive working environment of banking, the use of AI in banking operations is inevitable. The use and application of AI technologies helped banks to prevent fraud [12,13], have impactful operations [14], reliability and accuracy [14], high speed [15], and hassle-free banking services.

The banking sector in Pakistan is one of the most reliable and critical sectors, and it accounts for a significant portion of GDP contribution within the servicing sector of the economy. The application of AI technologies intervened and accelerated automation [16,17], blockchain [18,19], and Fintech [20,21]. Smartphone banking, ATMs, cash deposit machines, short message services, and emails all are integrated with AI technology in the present banking system in Pakistan. This integration has many uses, from data analysis to formulating strategies and setting future goals and objectives for banks.

In this dynamic era of AI, its role is increasing dynamically due to its quicker and faster response rate of processing information accurately from the database to respond in a competitive market [17,22]. AI is widely used in the areas of asset management, risk management, customer service, and data analysis. Moreover, given the nature of data in

the banks, AI has a significant role in processing data to predict the future of the economy and banking industry.

### 2.2. Key Areas of AI in Banking

*Cost saving*: The emergence of AI in the banking sector has significantly reduced the cost of paperwork and printing. According to [2], USD 416 billion will be saved by 2023 with the use of AI technology in the banking sector. The operating cost of the banks is to access information for managerial and customer use without incurring any personnel and paper costs.

*Chatbots*: Chatbot technology is one of the most unique and interesting AI technologies' software, which interacts with customers with preprogrammed queries of the customers for courteous, effective communication, and instant problem resolution [23]. Chatbot technology in the banks not only resolves the queries of the customers without human interaction but also collects data on customer queries, which can be used to resolve future problems [24].

*Customer experience*: Customer satisfaction and experience are proportionate to the adoption and use of digital financial services in banks. Customer preferences over the years have changed drastically and they demand quick responses with a personalized content. AI technology with machine learning uses a specific algorithm where banks can analyze and predict customer behavior and credit scores to develop customized plans for their customers [2]. AI can help banks to digitize their processes to meet customer expectations. The study on a sample of 360 banking customers from China revealed that perceived intelligence and perceived anthropomorphism have a significant positive impact on consumer's social support. This study exposed how AI affects consumers' satisfaction [25].

*Sentiments analysis*: The behavioral predictions of the customers is the core concern of any financial institution to develop and offer financial products and or services. The sentiment analysis technology of AI predicts customer emotions, feelings, and responses via emails, social media, and surveys to predict the preferences of the customers [23]. This technology collects information to develop and display the contents of the users given their preferences and choices [26].

*Automation*: The use of AI technology in the banking sector without human intervention can also be seen where digital machines count the currency accurately and quickly. This automation technology support increases the daily business volume of the banks, reducing work stress and the mathematical count error of cash-counting simultaneously. The use of automation systems in the banking sector has created a conducive working environment for the adoption of this technology in almost all the functional areas of financial institutions in the future.

*Fraud detection*: Financial institutions are more often exposed to the risk of fraud due to the large volume of business financial transactions and the complexity of the work tasks. As stated earlier, AI uses mathematical computation and complex algorithms that help monitor both customer and personnel behavior by implementing unsupervised learning programs [26,27]. Thus, fraud prevention with the use of AI technology can become easy [7]. AI is purely based on the machine learning programming approach to alternate human tasks within the banking sectors to avoid potential threats to business function performance.

### 2.3. Status of AI Use in Various Countries

Artificial intelligence and behavioral studies are of major interest to researchers for numerous reasons in the current era of business and technology. AI was first introduced to replace human efforts, and later was improved to recognize human working patterns and predict their behaviors. The banking industry in Pakistan, over the past two decades, has witnessed tremendous growth in the adoption of technology in its financial services. The growth of the financial and banking industry in Pakistan was recorded at 12.2% [6], and a 7% growth rate in the banking industry was recorded in the Kingdom of Saudi Arabia [28].

The offerings of financial services in the banking sector are focused on using information technology, keeping in view the customers' changing preferences. In this study, we used the theory of planned behavior (TPB) to evaluate the effect of the adoption of AI-based banking services by customers. The assessment of the behavioral tendencies on the adoption of banking services by the customers is focused on awareness, attitude towards AI, subjective norms, perceived risk, perceived usefulness, knowledge of AI technology, and intentions to adopt AI in banking.

*Awareness and adoption intentions*: The adoption and use of digital financial services are closely linked with the level of awareness of the customer towards the financial offerings of the banks. According to [29], customer awareness, initial trust, compatibility, and perceived risk are proportionate to the adoption of digital financial services. Ref. [30] revealed that the customer's ability to understand and use AI-based financial services tends to have a positive effect on the adoption of digital banking services. Thus, it is clear that higher customer awareness of AI digital services by the customer and their ability to understand the benefits and usage leads to a direct positive effect on the adoption intentions of banking services. A study was conducted on the use of AI on a sample of 400 banking customers in Thailand. It was found that the influencing factors were trust, social norm, perceived usefulness, and knowledge of using applications [31]. Another study was conducted to provide a model of customer churn prediction for the retail customers of a commercial bank in Iran. By applying advanced data analysis techniques on the transaction and operation data of the bank's customers, a well-designed classification of customers was presented for the churn rate. The customer churn prediction was performed by using artificial neural network [32].

*Attitude and adoption toward AI banking*: The first independent construct of TPB is the attitude, which is the sum value of feelings and emotions towards the object of interest [33,34]. AI is an innovative method of offering financial services to customers. According to [34], attitudes positively influenced the adoption of internet banking services in the UAE. In settings of AI-based financial offerings, the customer information and image of the banking services must be well incorporated to increase the adoption intentions.

*Subjective norms and adoption intentions of AI banking*: Subjective norms refer to the behavior of the individuals who either support, approve or disapprove of a particular behavior [35]. The empirical studies have established that the adoption of AI banking services is closely linked with the opinion of an influential group of people [36,37]. The usage and adoption of AI-based financial services are rising over time, particularly after the COVID-19 pandemic, when social distancing became an important measure to prevent the virus [38]. found that a group of influential people using AI services tends to have a positive effect on the adoption.

*Perceived risk and adoption intentions of AI banking*: Perceived risk is often described as to cost of spirit associated with the customer's buying behavior toward the object of interest [38,39]. AI is an innovative mode financial services used over the internet, which works with mathematical algorithms. The studies have found that the higher the perceived risk of customers toward AI-based financial offerings, the lower their adoption intentions [40,41]. In Pakistan, AI-based financial offerings are in very early stages, and hence, require customer trust and consistency regarding the outcome of the financial transaction to mitigate the risk cost of the customers. Moreover, the studies have also found that the infrastructure of AI-based technology has a significant influence on developing customer trust.

*Perceived usefulness and adoption of AI banking*: Perceived usefulness refers to the extent to which the use of the system will result to an enhanced performance [42,43]. AI is an alternative to human efforts, which not only improves the speed of transactions but also improves the working performance with more accurate information. The studies have established that the intervention of technology and the internet of things made banking self-dependent [44,45]. Individual performance and data security over the internet while

performing transactions increases, which leads to increased customer trust and in turn increases customer adoption of AI banking.

*Knowledge of artificial technology and adoption of AI banking*: Knowledge of artificial technology is one of the most influential factors that determine customer adoption intentions toward AI banking. According to [46], knowledge on information technology increases customer trust, and thus, leads to having a positive effect on the adoption of AI-technology-based financial services. The studies have also found that knowledge of financial services and their operations increases customer tendency since the customer already possesses the basic information [44]. The adoption of E-services in the financial industry and their adoption in banks are closely related [47]. Thus, we formulated the following hypotheses for this independent variable.

*2.4. Hypotheses Development*

**Hypothesis 1 (H1):** *Awareness is proportionate to the intentions to adopt artificial intelligence in banking.*

**Hypothesis 2 (H2):** *Attitude toward AI is proportionate to the intentions to adopt artificial intelligence in Banking.*

**Hypothesis 3 (H3):** *Subjective norms are proportionate to the intentions to adopt artificial intelligence in banking.*

**Hypothesis 4 (H4):** *Perceived risk is negatively proportionate to the intentions to adopt artificial intelligence in banking.*

**Hypothesis 5 (H5):** *Perceived usefulness is proportionate to the intentions to adopt artificial intelligence in banking.*

**Hypothesis 6 (H6):** *Knowledge of artificial intelligence technology is proportionate to the intentions to adopt artificial intelligence in banking.*

## 3. Data and Methodology

*3.1. Theoretical Framework*

The purpose of this study is to investigate the effect of customer awareness, attitude, subjective norms, perceived risk, perceived usefulness, and knowledge of artificial intelligence technology regarding intentions to adopt artificial intelligence in banking. To fulfill this aim, a theoretical framework is proposed by using six independent and one dependent variable, as shown in Figure 1.

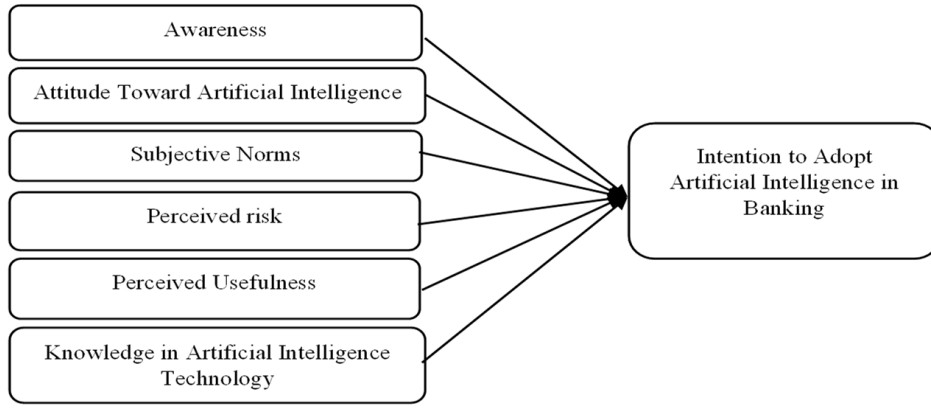

**Figure 1.** This is a figure. Schemes follow the same formatting.

The Table 1 shows the abbreviations of variables used in the study.

**Table 1.** Abbreviations of variables.

| Variables | Abbreviations |
|---|---|
| Awareness | AWR |
| Attitudes | ATT |
| Subjective norms | SN |
| Perceived risk | PR |
| Perceived usefulness | PU |
| Knowledge in artificial intelligence technology | KNG |
| Intentions to adopt artificial intelligence in banking | INT |

*3.2. Econometric Model*

The economic model used in this study is as follows:

$$\text{INT} = \beta_0 + \beta_1(\text{AWR})_i + \beta_2(\text{ATT})_i + \beta_3(\text{SN})_i + \beta_4(\text{PR})_i + \beta_5(\text{PU})_i + \beta_6(\text{KNG})_i + \text{\euro} \quad (1)$$

*3.3. Questionnaire Design and Sample*

The primary data are collected through a questionnaire based on a Likert scale ranging from 1, strongly disagree, to 5, strongly agree. The questionnaire is designed by using different studies, such as [48–53]. The questionnaire comprises seven demographic factors, which are age, gender, marital status, employment structure, educational level, and country. Along with seven demographic factors, the other seven variables are defined by using twenty-six items, four of which are related to awareness, perceived usefulness, perceived risk, subjective norms and knowledge in artificial intelligence technology. The other three items are related to the attitude towards AI and intention to adopt artificial intelligence in banking. Data are collected from five Asian countries due to time constraints; the five Asian countries are: Pakistan, China, Iran, Saudi Arabia, and Thailand. Therefore, a convenient sampling technique is used for the collection of data. A total of 799 customers participated in this survey. Empirical results are extracted to measure the reliability through Cronbach's Alpha, descriptive statistics, correlation, multiple regression, and one-way ANOVA by using the EVIEWS-12 and SPSS-24. Table 2 shows the response rate of the questionnaire.

**Table 2.** The response rate of questionnaires.

| Name of the Country | Response Received | Response Rate (%) |
|---|---|---|
| Pakistan | 300 | 37.5 |
| China | 111 | 14 |
| Iran | 110 | 13.7 |
| Saudi Arabia | 175 | 22 |
| Thailand | 103 | 12.8 |
| **Total** | **799** | **100** |

## 4. Results and Discussions

A total of 799 responses were collected for analysis purposes from the five Asian countries, which are Pakistan, China, Iran, Saudi Arabia, and Thailand. Before performing an in-depth analysis, it is very important to check the reliability of each of the items in the questionnaire. There is a total of seven items, of which six items are related to the independent variables and one item is related to the dependent variable. For further analysis, descriptive, correlation, multiple regression, and ANOVA will be performed.

*4.1. Reliability Analysis*

To determine the reliability, Cronbach's Alpha is used for all the items included in the questionnaire. If the value of Cronbach's Alpha is greater than 0.6, the variable is reliable and used for further analysis. Table 3 shows the reliability of each variable.

**Table 3.** Response rate of questionnaires.

| Variables | Cronbach's Alpha |
|---|---|
| Intention of adoption of artificial intelligence | 0.77 |
| Awareness | 0.65 |
| Attitude | 0.80 |
| Subjective norms | 0.70 |
| Perceived risk | 0.69 |
| Perceived usefulness | 0.90 |
| Knowledge of information technology | 0.75 |

According to the results, the variables show that the value of Cronbach's alpha is higher than the cutoff point, which is a sign that the items and the including variable are reliable and the results extracted by using these variables are also reliable and generalizable.

### 4.2. Demographics Profile

Seven demographic factors, namely age, gender, marital status, employment structure, educational level, and country, are used to describe the sample population used in this study. Table 4 represents the details of the demographic profile.

**Table 4.** Demographic profile.

| Demographics | Categories | Frequency | Percent |
|---|---|---|---|
| Age | 18–25 | 319 | 40 |
| | 26–35 | 189 | 24 |
| | 36–45 | 138 | 17 |
| | 46–55 | 67 | 8 |
| | 56–65 | 52 | 6.5 |
| | 66 and above | 34 | 4 |
| Gender | Male | 366 | 46 |
| | Female | 433 | 54 |
| Marital status | Single | 490 | 61 |
| | Married | 309 | 39 |
| Employment structure | Full-time salaried | 248 | 31 |
| | Part-time salaried | 98 | 12 |
| | Self-employed | 76 | 9.5 |
| | Unemployed | 197 | 24.5 |
| | Retired | 53 | 6.6 |
| | Housewife | 42 | 5.8 |
| | Student | 85 | 10.6 |
| Educational level | Lower than a high school | 64 | 8 |
| | High school or equivalent | 126 | 15.8 |
| | Diploma/advanced diploma | 67 | 8.5 |
| | Bachelor's degree | 176 | 22 |
| | Master's degree | 242 | 30.2 |
| | Ph.D. degree | 124 | 15.5 |
| Country | Pakistan | 300 | 37.5 |
| | China | 111 | 14 |
| | Iran | 110 | 13.7 |
| | Saudi Arabia | 175 | 22 |
| | Thailand | 103 | 12.8 |

A total of 799 respondent belong to the five Asian countries, which are Pakistan, China, Iran, Saudi Arabia, and Thailand. Among these, 46% are males and 54% are females. A total of 40% of respondents are in the age bracket of 18 to 25 years old. This means that the respondents have limited age-related experience when interacting with the banking system.

Most probably, the people in this age group belong to the group of students or early careers. Likewise, 31% of the respondents who participated in the survey have full-time salaried jobs. As for the educational level, 30.2% of respondents have a master's degree in various fields. It is observed in the demographics that highly educated and settled people from these five Asian countries participated in the survey related to the adoption of artificial intelligence in the banking sector.

### 4.3. Descriptive Analysis

To determine the true picture of the data, it is important to perform a descriptive analysis of each variable included in the study.

Descriptive statistics show the deviations of values from their mean values. Table 5 shows the descriptive statistics of all variables, including dependent and independent variables. It is observed that there are fewer deviations and spread shown by the variables, thus, the values are mostly near to their means.

**Table 5.** Descriptive statistics.

| Variables | Mean | Median | Mode | Standard Deviation | Minimum | Maximum |
| --- | --- | --- | --- | --- | --- | --- |
| AWR | 3.14 | 3.12 | 3.12 | 0.49 | 1.75 | 4.37 |
| PU | 3.51 | 3.5 | 4 | 0.81 | 1.25 | 5 |
| PR | 3.18 | 3.25 | 3 | 0.76 | 1.25 | 5 |
| SN | 3.36 | 3.5 | 4 | 0.68 | 1 | 5 |
| ATT | 3.51 | 3.66 | 4 | 0.81 | 1 | 5 |
| KNG | 3.45 | 3.5 | 4 | 0.79 | 1 | 5 |
| INT | 3.39 | 3.33 | 3 | 0.82 | 1 | 5 |

Note(s): AWR = awareness, PU = perceived usefulness, PR = perceived risk, SN = subjective norms, ATT = attitude towards artificial intelligence, KNG = knowledge in artificial intelligence technology, INT = intentions to adopt artificial intelligence in banking sector.

### 4.4. Correlation Analysis

Correlation is the statistical technique that is used to determine the relationship or association between two variables. The value of the coefficient of correlation ranges from +1 to −1. The direction of the relationship is indicated by the sign of the coefficient; a positive sign indicates a positive relationship, and a negative sign indicates a negative relationship. If the value of the coefficient of correlation is close to +1, this means that the variable is positively correlated; likewise, if the value of the coefficient of correlation is close to −1, the variables are said to be negatively correlated. Table 6 shows the correlation among the variables. Usually, we are interested in observing the relation between the independent and independent variables. Table 6 represents the correlation matrix.

In this study, INT shows a strong correlation with PU and ATT, and a moderately strong correlation with AWR, SN, and KNG, while INT shows a weak correlation with PR. It is also observed that all variables are significantly and positively correlated with each other.

### 4.5. Regression Analysis

Regression analysis is an important technique that is used to determine the effect of different variables that are known to be independent variables on the dependent variable. In this study, multiple regressions are applied, as there are more than one independent variable and a single dependent variable.

Table 7 shows the impact of awareness, attitude towards artificial intelligence, perceived risk, perceived usefulness, subjective norms, and knowledge of artificial intelligence on the intention to adopt artificial intelligence in banking. According to the F-statistics, the overall model is significant at $p < 0.01$. This means that if the model is valid, the results

generated by using this model also are reliable and generalizable. R and adjusted R square give the explanatory power of the independent variables, which is how much the dependent variable is explained by the independent variable. In this case, approx. 40% of the dependent variable is explained by the independent variables. The first hypothesis of the study states that awareness is proportionate to the intentions to adopt artificial intelligence in banking. The results for $H_1$ show a positive and highly significant relationship between awareness and intentions to adopt artificial intelligence in banking with a $p < 0.01$. The second hypothesis is about the relationship between the attitude toward artificial intelligence and intentions to adopt artificial intelligence in banking. The results for $H_2$ were also found highly significant and positive with a $p < 0.01$. This means that if people have a positive attitude toward artificial intelligence, they most likely intend to adopt artificial intelligence in banking. Likewise, the third hypothesis of the study is about the relationship between subjective norms and intentions to adopt artificial intelligence in banking. The result for the relationship between the subjective norms and intentions is positive and highly significant with a $p < 0.01$.

**Table 6.** Correlation matrix.

| | AWR | PU | PR | SN | ATT | KNG | INT |
|---|---|---|---|---|---|---|---|
| **AWR** | 1 | | | | | | |
| **PU** | 0.190606 | 1 | | | | | |
| **PR** | 0.031843 | 0.091481 | 1 | | | | |
| **SN** | 0.181903 | 0.229269 | 0.12912 | 1 | | | |
| **ATT** | 0.129814 | 0.407974 | 0.07274 | 0.356612 | 1 | | |
| **KNG** | 0.183954 | 0.35195 | 0.07489 | 0.315205 | 0.39677 | 1 | |
| **INT** | 0.239083 | 0.452354 | 0.03247 | 0.354279 | 0.53528 | 0.397188 | 1 |

Note(s): AWR = awareness, PU = perceived usefulness, PR = perceived risk, SN = subjective norms, ATT = attitude towards artificial intelligence, KNG = knowledge in artificial intelligence technology, INT = intentions to adopt artificial intelligence in banking sector.

**Table 7.** Regression analysis.

| Variables | Coefficient | Std. Error | t-Statistic | Prob. |
|---|---|---|---|---|
| AWR | 0.179224 | 0.047158 | 3.800507 | 0.0002 *** |
| ATT | 0.339769 | 0.032784 | 10.36395 | 0.0000 *** |
| PR | −0.044788 | 0.029748 | −2.505600 | 0.0003 *** |
| PU | 0.224927 | 0.031342 | 7.176467 | 0.0000 *** |
| SN | 0.153697 | 0.036396 | 4.222868 | 0.0000 *** |
| KNG | 0.131906 | 0.032490 | 4.059865 | 0.0001 *** |
| R-squared | 0.401954 | Mean dependent var | | 3.397547 |
| Adjusted R-squared | 0.397424 | S.D. dependent var | | 0.822959 |
| F-statistic | 88.71891 | Durbin-Watson stat | | 2.166865 |
| Prob(F-statistic) | 0.000000 | | | |

Note(s): *** $p < 0.01$. AWR = awareness, PU = perceived usefulness, PR = perceived risk, SN = subjective norms, ATT = attitude towards artificial intelligence, KNG = knowledge in artificial intelligence technology.

The fourth hypothesis of the study is about the negative relationship between the perceived risk and intentions to adopt artificial intelligence in banking. The results for $H_4$ show the negative and significant relationship between the perceived risk and intentions of adoption of artificial intelligence in banking with a $p < 0.01$. The fifth and sixth hypotheses of the study are about the relationship between the perceived usefulness and knowledge of artificial intelligence technology and the intentions to adopt artificial intelligence in

banking. The results found for these two hypotheses are positive and highly significant, with a *p* < 0.01. Table 8 shows the summary of the hypotheses testing extracted from the regression analysis.

**Table 8.** Summary of the hypotheses testing results from regression analysis.

| Hypotheses | Remarks |
|---|---|
| **Hypothesis 1** | Accepted |
| **Hypothesis 2** | Accepted |
| **Hypothesis 3** | Accepted |
| **Hypothesis 4** | Accepted |
| **Hypothesis 5** | Accepted |
| **Hypothesis 6** | Accepted |

*4.6. ANOVA Analysis*

ANOVA analysis was used to find whether there is any difference in the intentions of adoption of artificial intelligence between the countries and at different levels of education. Table 9 describes the F-stats for the variable country and education level by taking the intentions of the adoption of artificial intelligence in banking.

**Table 9.** ANOVA analysis of intentions of adoption of artificial intelligence in banking.

| Variables | Mean Square | F Value | Pr > F |
|---|---|---|---|
| Country | 24.77541483 | 45.11 | <0.0001 |
| Education | 3.02 | 3.02 | 0.0105 |

The significant F-value indicates that there is a difference in the people regarding the intentions of adopting artificial intelligence in banking across the countries and at different levels of education. For more clarity, grand means are calculated for both country and education. Tables 10 and 11 show the grand means for country and education level.

**Table 10.** Summary of grand means by country.

| Country | Grand Means |
|---|---|
| Pakistan | 3.67782589 |
| China | 2.92607570 |
| Iran | 3.14941793 |
| Saudi Arabia | 3.57831986 |
| Thailand | 2.92249128 |

**Table 11.** Summary of grand means educational level.

| Educational Level | Grand Means |
|---|---|
| Lower than a high school | 3.29984993 |
| High school or equivalent | 3.12933670 |
| Diploma/advanced diploma | 3.08036760 |
| Bachelor's degree | 3.23596819 |
| Master's degree | 3.31105351 |
| Ph.D. degree | 3.44838085 |

According to the results in Table 10, the mean score table indicates that the mean score of Pakistan is 3.67, and for China, it is 2.92, followed by Iran with a mean of 3.14, Saudi Arabia with 3.47, and Thailand is 2.92. This mean score shows that the people's intentions for the adoption of artificial intelligence in the banking sector widely vary from one another. It is also shown in Figure 2 graphically.

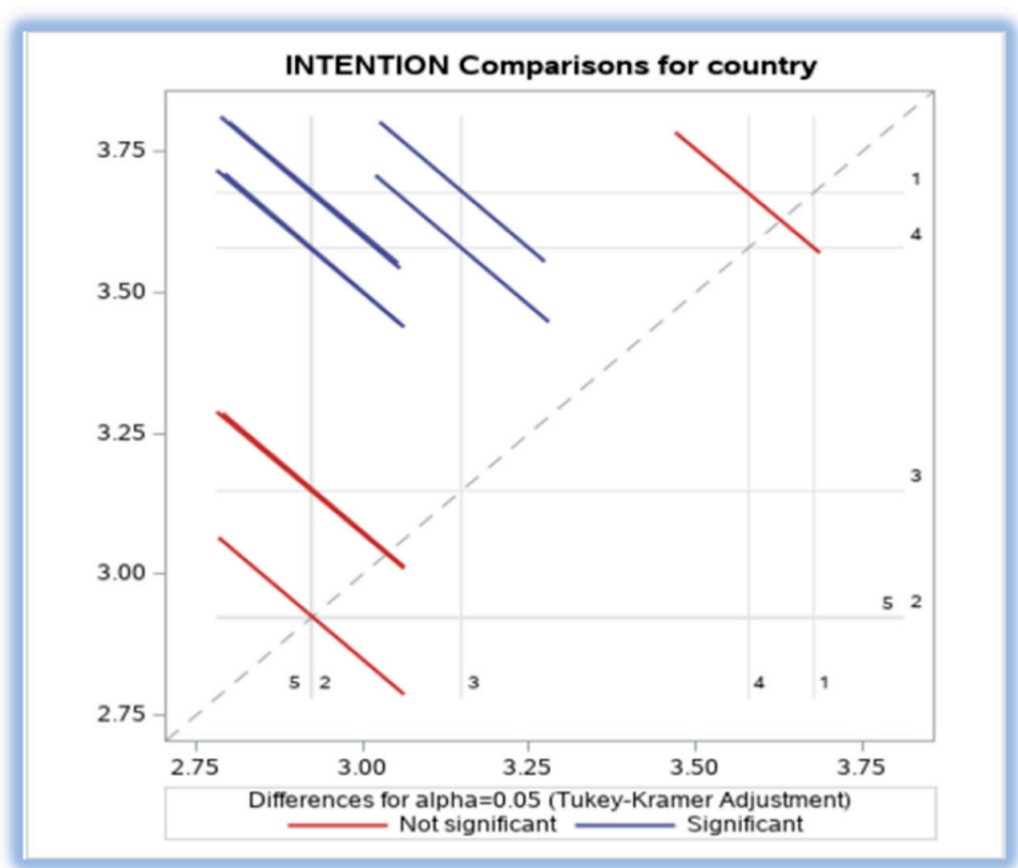

**Figure 2.** Intentions to adopt artificial intelligence, comparisons for country.

Table 11 shows that the result obtained by analyzing the intentions of people to adopt artificial intelligence in the banking sector across different levels of education, show that people's intentions vary at different educational levels, with the means value of lower than high school being 3.29, high school being equal to 3.12, diploma being 3.08, bachelor's degree, 3.23, master's degree, 3.31, and Ph.D. degree being 3.44. This is also represented in Figure 3 graphically.

*4.7. Discussions*

Artificial intelligence has acquired an important place in today's world in every field of life. Developed and developing countries are also trying to incorporate artificial intelligence in different sectors. Among these, banking is one of the important sectors for any country. With the speedy advancement in technology, artificial intelligence plays a crucial role for individuals, as well as organizations, especially in the banking sector where there are risks of fraud and speedy transactions are required. This study is based on five Asian countries: Pakistan, China, Iran, Saudi Arabia, and Thailand, whereby the study tries to determine the intentions of people regarding adopting artificial intelligence in the banking sector, concerning awareness, attitude, subjective norms, perceived risk, perceived usefulness and knowledge of artificial intelligence technology.

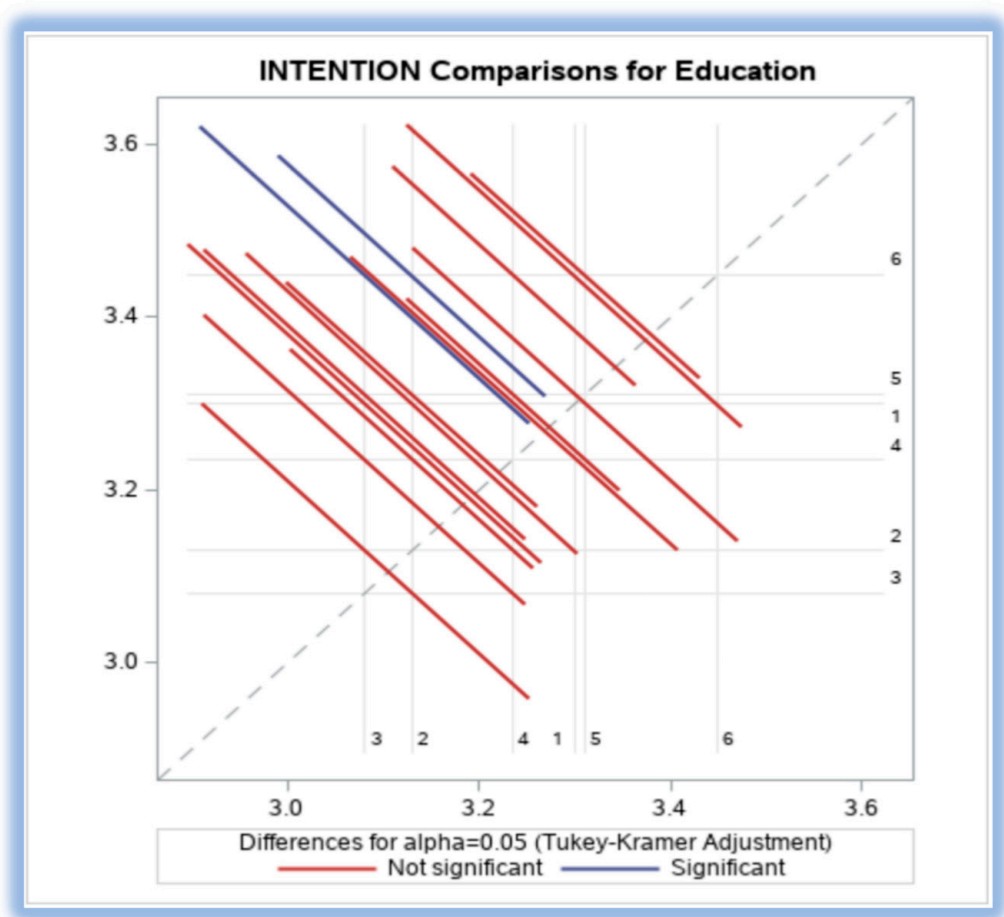

**Figure 3.** Intentions to adopt artificial intelligence. Comparisons for educational level.

The results of the first hypothesis are that there is a positive relationship between awareness and intentions to adopt artificial intelligence. They depict that if people's or customers' awareness of digital financial services leads to customer trust, and people are more likely to adopt artificial intelligence in banking. The results are consistent with [30], which revealed that the customers' ability to understand and use AI-based financial services tends to have a positive effect on the adoption of digital banking services. Likewise, the second, third, fifth, and sixth hypotheses are also accepted by showing a significant and positive relationship. The attitude, subjective norms, perceived usefulness, and knowledge of artificial intelligence also show a positive relationship with the intentions to adopt AI in banking. These results are also confirmed by researchers such as [34,38,44,47].

However, the fourth hypothesis is about the negative and significant results between risk perception and intentions to adopt artificial intelligence in banking. This means that the people or the customer perceive risk in using the digital services of the bank. This may be due to the lack of knowledge or trust and being unable to mitigate the financial cost associated with the use of digital technology in the banking sector. To overcome the perceived risk, it is important to build the trust of the customer by using artificial intelligence in banking. The results are consistent with the results in [40,41].

Generally, it is also observed that people are not ready to take risks, especially in financial matters. Furthermore, the perception of risk in males and females differs, as the females are more risk-averse than the males. Another important aspect is that males are more involved in financial matters. Artificial intelligence is a new technology and very few people in developing countries know about it. Due to this, people hesitate to adopt it. This hesitation in the adoption of artificial intelligence will be reduced by spreading awareness of using artificial intelligence.

The ANOVA analysis also depicts the significant difference in the intentions of adopting AI related to countries and educational levels. All customers using banking services living in Pakistan, China, Iran, Saudi Arabia, and Thailand show a difference in their intentions of using AI in the banking sector. The reason behind this is the difference in the level of growth in these Asian countries. Pakistan is an underdeveloped country and does not have many resources to update the technology or to take the necessary steps to make customers aware of the usage of digital financial services. The other four countries, China, Iran, Saudi Arabia, and Thailand are more developed than Pakistan and they introduced artificial intelligence in their banking sector earlier than Pakistan. Likewise, the difference in educational level also shows the difference in the intention to adopt AI in banking. This means that the people or the customers who are highly educated are more ready to adopt AI in their banking operations than the customers who have low educational levels.

## 5. Conclusions

The transformation of the banking sector was not an overnight process. It went through traditional banking (1472) to AI-based banking (2017 onwards), which is a major shift in the banking sector; this change will be observed more quickly in banking areas like core banking, operational performance, and customer service. The purpose of this study is to comprehend the difficulties associated with integrating AI and the customer's intentions toward AI adoption in the banking industry. Therefore, this study used the exploratory approach by using the quantitive research design to examine the relationship between the independent variables/predictors (AWR, ATT, SN, PR, PU, and KNG) and the dependent variable INT to adopt AI in the banking sector.

The result shows that all predictors except PR had a significant and positive relationship between INT and the intention to adopt AI in the banking industry, whereas PR shows a negative but significant relationship between INT and the intention to adopt AI. In addition, the analysis also revealed that there is a difference in the adoption of AI in the banking sector regarding the country and educational level of customers. The study contributes to the existing literature related to artificial intelligence by collecting data from five Asian countries (Pakistan, China, Iran, Saudi Arabia, and Thailand). This indicates that it is a comprehensive study comprising five Asian countries to view the trends of AI in the banking sector. In addition, the findings give worthy insights for the creation of future strategies by the banking management, such as AI algorithms, which can gather and analyze consumer data, offer pertinent product recommendations that have already been approved, and offer individualized financial guidance. AI in banking apps can make keeping track of financial objectives and outgoings simple for customers.

This study gives practical implications and recommendations to banking management, policymakers, government, and technological regulatory bodies. The results of this study can help the banking management to update and revise their marketing strategy of building or increasing the trust of customers, which helps them to overcome the risk of using digital technology while making transactions. Furthermore, this study recommends that bank managements and technology regulatory authorities take the required actions to improve security and protection measures that guarantee improved customer service to increase the reliability and appeal of AI in banking service.

Some of the aspects of this study need further research. This study was limited to five Asian countries. Further research may be conducted on other Asian and European countries and compare results by analyzing the perspective of customers regarding artificial intelligence. In this study, only the banking sector was targeted. In the future, other financial sectors will also be considered for further study. From a methodological standpoint, this study depends on a survey to obtain self-reported data. Such data may not be very precise. In the future, to increase accuracy, it is suggested that researchers increase the sample size and adopt other methods of data collection, such as field experiments. In this study, due to time constraints, mediation and moderation were skipped; future research will introduce the mediating moderating variables that have a strong impact on AI by updating the

model. Once AI technology has been broadly incorporated into the business environment, the findings of this study may vary over time. Therefore, more research is required to determine the changes in consumer intention between early adopters and late adopters of AI in the banking industry.

**Author Contributions:** Conceptualization, U.N. and M.A.; Methodology, A.S.; Formal analysis, A.S.; Investigation, A.S.; Resources, U.N. and Z.A.; Data curation, Z.A. and M.A.; Writing—original draft, U.N., A.S., Z.A. and M.A.; Writing—review and editing, U.N. All authors have read and agreed to the published version of the manuscript.

**Funding:** The authors would like to acknowledge the support of the Research & Initiative Centre (RIC), Prince Sultan University for paying the article processing charges (APC) of this publication.

**Institutional Review Board Statement:** Not applicable.

**Informed Consent Statement:** Not applicable.

**Data Availability Statement:** Not applicable.

**Conflicts of Interest:** The authors declare no conflict of interest.

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
