# Peer review of "Banking 4.0: Artificial Intelligence (AI) in Banking Industry & Consumer’s Perspective"

_sustainability, doi:10.3390/su15043682_

Round 1

Reviewer 1 Report

The article "Banking 4.0: Artificial Intelligence (AI) in Banking Industry & Consumer’s Perspective" presents a study that aims to discuss the consumer’s perspective on artificial intelligence adoption in the bank sector in Asian countries. In this reviewer’s opinion, the paper needs improvements:

1) In the Abstract, insert the description of "INT" term.

2) There are many sentences "Artificial intelligence (AI)" in the text. Please, use the complete sentence once time and in the other occurs or use "artificial intelligence" or "AI".

3) In page 10 shows the term "ANOVA". What is ANOVA?

Author Response

Dear Reviewer,

Greetings! Please check the revised version and give approval for publication.

Thanks,

Kind regards,

Dr.Attayah Shafique

Reviewer 2 Report

First of all, I would like to state that the paper is well written, and very well based by updated bibliographical references and it presents a current and very interesting topic.

It should be referred in the article in more detail and more discussed the questionnaire, and the reasons of the questions maded.

To better analyze the simulation results, the figures must be more quality and/or more zoom. Also could be better described and justified the real advantage of this study.

On the other hand, also it should be presented and justified the informatics tools that were used.

As final conclusion, due to the overall scientific quality of the paper, I recommend that it is accepts for publication in this Journal but after changes.

Author Response

(The authors gave the same response as above.)

Reviewer 3 Report

The article is well written. The hypotheses are clearly presented. But I would like to see in the text an explanation of the Bank 4.0
concept that appears in the title of the article. I did not find it.

In the article there is a list of key areas for the use of AI in banking.
One such area is Automation. I agree that automation reduces the workload
and chance of money counting errors. But for a bank to adopt digital,
electronic and mechanical automation in its systems, it does not need to
use AI.

And going back to the list of key areas, only cost saving is directly
linked to the journal's topic: sustainability.
The framework proposed in figure 1 was very summarized.

The authors talk about behavioral trends about the adoption of AI services
in banks, but when they present examples of these behaviors, the examples
are actually about the adoption of digital services, services that do not
necessarily use AI algorithms.

Author Response

(The authors gave the same response as above.)

Reviewer 4 Report

This manuscript is “Banking 4.0: Artificial Intelligence (AI) in Banking Industry & Consumer’s Perspective”. Some detailed comments are as follows:

(1) This paper needs to use the template of the journal. The author can download the template on the website of the journal and modify it as required.

 (2) Page 1: The abstract of the paper lacks the disclosure of innovative content.

 (3) Page 2: The introduction chapter of the manuscript lacks the development analysis of “Artifical Intelligence (AI)”.

 (4) Pages 4-8: The “Key Areas of AI in Banking” and “Status of AI use in Various Countries” was analyzed, However, there is no comparative analysis of different countries.

 (5) Pages 18-20: The author only discusses with very little work.

 (6) Pages 20-21: The conclusion is only a general view, and there is no specific research conclusion.

For example:

1) Artificial intelligence (AI) is causing a major shift in the banking sector and this change would be observed more quickly than ever.

What do major shift mean?

2) The study contributes to the existing literature related to artificial intelligence by collecting data from five Asian countries (Pakistan, China, Iran, Saudi Arabia, and Thailand).

What contribution does the author mean?

3) In addition, the findings give worthy insights for making future strategies by the banking management.

What are worthy insights?

 (7) The format of references does not meet the requirements of the journal.

 (8) The quality of all Figs in the paper needs to be improved.

 (9) The format of the table does not meet the requirements of the journal.

 (10) A proof reading by a native English speaker should be carefully conducted to improve both language and organization quality.

For example:

1)Page 1: Improper abbreviations: Artificial intelligence (AI) is repeated many times.

2) Page 1: The “INT” is the abbreviation directly used, and the full name is not provided.

3) Page 4: The “CDM” and “SMS” appear only once and do not need to be abbreviated.

4) Pages 9 and 10: The tense is misused. “The primary data is collected through a questionnaire based on Likert scaling ranging …”, “The economic model used in this study is as follows:” etc.

There are many other similar mistakes.

Author Response

(The authors gave the same response as above.)

Round 2

Reviewer 3 Report

I would like to suggest to the authors the inclusion of the keyword Consumption change. This keyword is one of the Sustainability Journal subjects and the word Consumer are presented in the Title. The keyword artificial intelligence is too broad in this context and should be avoided.

In the section 2.2 (Key Areas of AI in Banking), when the authors talk about automation, I agree that automation reduces the workload and chance of money counting errors. But for a bank to adopt digital, electronic and mechanical automation in its systems, it does not need to use AI. I think these use of AI end bank service automation could be more detailed.

Author Response

Dear Reviewer,

Thank you for reviewing and commenting on the article. We have responded to all the valuable comments in the best way.

kind regards,

Dr.Attayah Shafique

Reviewer 4 Report

This manuscript is “Banking 4.0: Artificial Intelligence (AI) in Banking Industry & Consumer’s Perspective”. Some detailed comments are as follows:

(1) L139-204: “Status of AI use in Various Countries” was analyzed. The author only mentioned Pakistan and Saudi Arabia. Can these two regions represent all countries? The author needs to explain the representativeness of the selected region.

(2) Pages 13-14: The content of the discussion section still needs to be strengthened.

(3) The format of references does not meet the requirements of the journal. Authors can refer to published papers and revise them according to the requirements of the journal.

For example:

1) The page number of the journal in the reference should be provided.

2) The journal names in the references need to be abbreviated as required.

(4) The format of the table does not meet the requirements of the journal.

(5) The author has not completely revised the language of the manuscript. I suggest that the author entrust the polishing organization to polish the language.

For example:

1) L12,14: Improper abbreviations: Artificial intelligence (AI) is repeated many times. The “INT” is only used once, which does not require abbreviations.

2) L90: The “CDM” and “SMS” appear only once and do not need to be abbreviated.

L38, 89: The full name of “ATM” shall be provided for the first use.

There are also some problems in the use of abbreviations in the main body of the manuscript.

3) L431: The “Conclusion” should be changed to “Conclusions”.

There are many other similar mistakes.

Author Response

(The authors gave the same response as above.)

Round 3

Reviewer 4 Report

This manuscript is “Banking 4.0: Artificial Intelligence (AI) in Banking Industry & Consumer’s Perspective”. Some detailed comments are as follows:

The reviewer has mentioned it many times, but the author does not revise it. Why should the author repeat it twice about “artificial intelligence (AI)” ? (L12 and L14). The full name is used for the first time, while the full name is not required for the second time.

There are many other similar mistakes.

Author Response

Dear Reviewer,

Thank you so much for your comments and suggestions.

I revised the manuscript according to the suggestions. 

Regards,

Dr.Attayah Shafique

corresponding Author
